# The Effect of Physical Exercise and Internet Use on Youth Subjective Well-Being—The Mediating Role of Life Satisfaction and the Moderating Effect of Social Mentality

**DOI:** 10.3390/ijerph191811201

**Published:** 2022-09-06

**Authors:** Baole Tao, Hanwen Chen, Tianci Lu, Jun Yan

**Affiliations:** College of Physical Education, Yangzhou University, Yangzhou 225127, China

**Keywords:** youth, subjective well-being, physical activity, internet use, social mindset

## Abstract

Youth subjective well-being is enhanced not only from physical exercise but also from internet use. Based on the 2017 China General Social Survey (CGSS) data, the Bootstrap method was used to examine the mechanisms of the effects of physical exercise and internet use on youth subjective well-being. In this study, the questionnaire data of 619 Chinese young people (18–35 years old) were selected as the sample source. It was found that physical exercise (2.881 ± 1.352) and internet use (4.544 ± 0.756) had positive effects on youth subjective well-being (88.762 ± 11.793). Life satisfaction (2.253 ± 0.826) partially mediated the development of physical exercise and internet use on subjective well-being, with indirect effects of 34.1% and 30.4%, respectively. A social mindset (10.181 ± 1.966) played a moderating role in the relationship between physical exercise and youth subjective well-being and internet use and youth subjective well-being in both groups. The positive effects of physical exercise and internet use on youth subjective well-being gradually increased with the improvement in social mindset. This study revealed the mechanisms of physical activity and internet use on subjective well-being and that life satisfaction and the social mindset of youth are essential factors influencing subjective well-being.

## 1. Introduction

Subjective well-being is an individual’s overall cognition and evaluation of the quality of life, defined as their satisfaction with their life status and subjective perception of the consistency of all aspects of their existing life with their expectations [1]. SWB includes good material life, the identity relationship between the individual and the group, and the individual’s mental state. Therefore, SWB reflects the unity, harmony, and stability of an individual’s expectation of themselves and their actual life [2]. With the improvement in subjective well-being measurement and analysis techniques, the influencing factors of subjective well-being explored by the academic community have expanded from focusing on economic factors such as (absolute/relative) income and economic inequality (inequality between the rich and the poor) to personality characteristics, social demographics, and social, psychological, and political factors such as structure and social capital [3].

However, as two different lifestyles, there is still room for exploring the impact of physical exercise and internet use on subjective well-being. Nowadays, the internet has become an indispensable part of people’s life. The internet is widely used in real life. For the youth group, more and more young people will choose to use the internet to chat, play games, work, and live. However, internet use also hurts individual socialization. Long-term use of the internet will reduce the communication between people, and it will also be accompanied by hostile and violent social events, resulting in adverse social effects.

Similarly, physical exercise has always been a part of social life from its birth to the present, and it is closely related to human survival challenges, religious beliefs, physical education, political activities, and recreation. Active participation in physical exercise has many benefits for physical and mental health, enhancing health and enabling a better quality of life. People who regularly participate in sports competitions have a more comprehensive and systematic understanding of the relationship between people and teams and between people and others and are better at dialectically dealing with problems such as competition and cooperation. Modern society is pluralistic; the lifestyle has also changed from single to pluralistic. Physical exercise and internet use are coming into our lives across the board. Especially for young people, physical exercise and internet use have become an essential part of a modern lifestyle, contributing not only to physical and mental health but also to the health of society [4]. Existing studies have found that physical exercise and internet use impact subjective well-being [5,6]. At the same time, compared with other age groups, youth groups participate in physical exercise and use the internet more frequently [7,8]. Therefore, it is time to explore the impact of physical activity and internet use on the subjective well-being of youth groups.

Subjective well-being is the result of multiple social factors [9]. It is necessary to start from multiple dimensions to explore the influence mechanism of physical exercise and internet use on young people’s subjective well-being. Campbell pointed out that life satisfaction refers to people’s evaluation of their satisfaction with their living conditions according to their values and subjective preferences [10], which is the cognitive component of subjective well-being [11]. Therefore, life satisfaction is not only an individual’s satisfaction with their personal life but reflects an overall assessment of their quality of life. Furthermore, life satisfaction can reflect an individual’s subjective perception of the quantity and quality of social resources they possess. Therefore, personal subjective well-being is closely related to life satisfaction. In studying the influencing factors of subjective well-being, examining the influence of life satisfaction is of great significance.

On the one hand, for people who undertake physical exercise and use the internet, their life satisfaction may not be the same, and life satisfaction is related to subjective well-being. Whether physical exercise and internet use affect their subjective well-being through life satisfaction has become a topic worth exploring. On the other hand, promoting the improvement in youth’s subjective well-being is inseparable from an excellent social mentality. Social mentality refers to the macro, dynamic, and general social-psychology situation that diffuses in the whole society or a particular social group within a certain period. It covers the structure of social needs, cognition, emotions, values, and actions. It includes specific stable social-psychological characteristics and some temporary conditions, reflecting the macro-psychological relationship formed by the mutual construction between individuals and society [12]. At a deeper level, people with different social mentalities may have different degrees of change in life satisfaction and subjective well-being when they participate in physical exercise and use the internet. Therefore, at the micro-individual level, life satisfaction and social mentality may shape the association between physical exercise, internet use, and youth subjective well-being. Previous studies only explored the effects of physical exercise and internet use on life satisfaction and subjective well-being, but this study innovatively linked the two life behavior styles together and added social mentality as a moderating variable to better reveal the rules.

## 2. Literature Review and Research Hypotheses

### 2.1. Effects of Physical Activity and Internet Use on Subjective Well-Being

The most significant impact on individual subjective well-being is economic factors and lifestyle changes [13]. In other words, the effect of lifestyle on youth well-being is dynamic. Physical exercise and internet use are the most common behavioral manifestations in individual social life, which may impact youth well-being. Numerous studies have shown that physical activity and internet use are associated with subjective well-being. As an essential source of well-being, physical exercise is an individual’s subjective expectation and feeling of their own physical and mental changes. At the time, internet use is an expectation of future social mobility, which also affects individual well-being.

Physical exercise is an activity whose primary purpose is to enhance physical fitness and improve health through various sports methods and means according to the needs of individuals to maintain their physical and mental health. Physical exercise is a source of health and happiness. Individual participation in physical exercise can generate a sense of pleasure, enhancing their psychological well-being [14]. European-based social survey data shows a linear relationship between physical activity and subjective well-being. Higher physical activity levels are associated with higher subjective well-being scores [15]. Adolescent individuals are at an important stage of completing socialization and having a harmonious self. For people at this stage of development, physical exercise is likely to affect their subjective well-being by acting on social relations and self-evaluation [16].

With the development of the network information society, the internet has become an essential part of young people’s lives. An annual comparison shows that in the past five years, the internet and the subjective well-being of young people have generally shown an upward trend [17]. Internet use has a significant positive impact on subjective well-being. Social trust is an essential mechanism of internet use affecting youth subjective well-being. Internet use improves social trust, thereby increasing the incidence of subjective well-being [18]. To sum up, this paper makes the following assumptions:

**H1.** *Physical exercise has a positive effect on youth subjective well-being*.

**H2.** *Internet use has a positive effect on youth subjective well-being*.

### 2.2. Effects of Physical Activity and Internet Use on Subjective Well-Being: The Mediating Role of Life Satisfaction

It is not enough to identify the effects of physical activity and internet use on youth subjective well-being, and the underlying mechanisms need to be explored. In the current stage of economic development, young people’s sensitivity to economic indicators has declined, and they are particularly concerned about their satisfaction with various quality of life indicators such as education, medical care, and social security. As an important indicator, life satisfaction can enhance individuals’ awareness of a better life. Feelings of being alive and well thereby improve subjective well-being [19].

Generally, a person with high life satisfaction has higher well-being and positive emotional experiences and, therefore, less socially dissatisfied emotions and behaviors. In comparison, a person with low life satisfaction has lower well-being, and more negative emotional experiences are more likely to produce social deprivation and antisocial behavior. Overall, life satisfaction is conceptually close to subjective well-being and is an individual’s comprehensive experience evaluation of many areas of life (such as work, emotions, and income) [20]. Life satisfaction emphasizes the cognitive component of subjective well-being [21]. To sum up, life satisfaction can enhance the subjective well-being of young people. Therefore, we propose the following hypothesis:

**H3.** *Life satisfaction has a positive effect on subjective well-being*.

At the same time, life satisfaction is also affected by external social factors such as life events and lifestyle behaviors (such as physical exercise) [22]. Studies have shown that people who undertake more physical activity have higher life satisfaction [23]. Participating in physical activity was associated with a higher quality of life and life satisfaction than those who did not participate in physical activity [24]. Zheng’s research shows that the leisure satisfaction and well-being of leisure sports participants in China are positively correlated [25]. Therefore, physical exercise not only directly affects the subjective well-being of youth but also plays a positive role in subjective well-being by improving life satisfaction. In this regard, we propose the following hypotheses:

**H4.** *Physical exercise has a positive effect on life satisfaction*.

**H5.** *Life satisfaction mediates between physical activity and youth well-being*.

At present, the research results on the impact of internet use on well-being are quite rich, and there is a consensus on the view that internet use significantly affects well-being [26]. The use of media brings psychological and behavioral demands to people. There is a causal chain of “social factors + psychological factors—media expectations—media contact—demand satisfaction” [27]. As a newly developed network medium, the internet is becoming increasingly mature. Through the media role of the internet, young people can meet their needs for entertainment, leisure, learning, making friends, and personal development to improve subjective well-being. Moreover, the “network gain effect theory” posits that the internet is a new way and a necessary means for young people to maintain or establish new relationships. Internet technology will promote young people’s social participation, enhance their social capital, and help improve life satisfaction [28]. In this regard, this paper puts forward the following assumptions:

**H6.** *Internet use has a positive effect on life satisfaction*.

**H7.** *Life satisfaction mediates between internet use and youth subjective well-being*.

### 2.3. Effects of Physical Activity and Internet Use on Subjective Well-Being: The Moderating Role of Social Mentality

The social mentality is the mental state of the spiritual structure of the social subject. It is a system composed of three levels: social orientation, social reason, and spiritual pillar, and has the functions of intermediary filtering, two-way effect, and self-regulation. The social mentality is not only a reflection of social transformation but also a force affecting social transformation. It also directly affects individual actions [29]. Therefore, social mentality may play a moderating role in the relationship path of each variable in this study. Social mentality may affect the effect of physical activity and internet use on subjective well-being. Studies have begun to explore the mechanism of action in the relationship between social mentality and subjective well-being [30]. The study found that an excellent social mentality strengthens the positive effect of physical exercise and internet use on youth’s subjective well-being [31,32].

With the improvement in people’s quality of life, physical exercise has become an indispensable part of people’s daily life. People who regularly participate in leisure sports activities have higher life satisfaction. Although the research on the relationship between physical exercise and life satisfaction is vibrant, the research on the elderly and students accounts for the majority, and the young and middle-aged groups are ignored. For young people composed of the post-1980s–1990s age group, although they are the beneficiaries of the reform dividend, the resulting wrong social mentality should not be underestimated. More and more young people are labeled as “anxious,” “decadent,” and other negative labels. Because social mentality has a unique change process, it can bring specific social–psychological effects, such as the convergence effect, herd effect, and group polarization effect [33]. By adjusting according to this change process, the individual can form a peaceful state, maintain a neutral state, and avoid a malignant state. Therefore, a positive social mentality can regulate the subjective well-being of individuals. To sum up, we make the following assumptions:

**H8.** *Social mentality plays a moderating role between physical activity and youth subjective well-being*.

**H9.** *Social mentality plays a moderating role between physical activity and youth life satisfaction*.

In current academic circles, there are different opinions on the relationship between internet use and individual subjective well-being, which is both positive and negative. For example, it has been pointed out that internet use reduces an individual’s real-world activities and makes them more lonely, thus creating a negative relationship with an individual’s subjective well-being [34]. Some studies also point out that internet use can have a positive correlation with individual subjective well-being through social support [35]. The above research results suggest that there may be a deeper relationship between internet use and individual subjective well-being, and the direct relationship cannot be easily investigated. Since every individual lives in the society and is a part of the society, there may be a certain relationship between the individual’s social mentality and his subjective well-being and internet use. With the rapid development of network technology, new media and micro media have become a normal way for young people to study, live, work, play, and socialize. In such a big background, irrational social mentality will lead to the young people’s social mentality showing blind conformity, emotional susceptibility, pluralistic values, multiple contradictions, and stratified differences. When the irrational social mentality is stimulated and driven by strong emotions, reason and truth will fall into the whirlpool of emotions. Group and individual psychology and expectations become the driving factors for netizens to ignore facts and go with the flow. In many popular online events, the competition of different emotional camps, the catharsis of emotions, and the emergence of various versions of rumors are all manifestations of the spread of irrational mentality driven by negative emotions. Therefore, irrational negative social mentality will lead to problematic internet use and problematic network behaviors. In addition, in the process of internet use, irrational and negative social mentality will also have a certain negative relationship with the ability of individuals to filter negative information, which may lead to the decline in subjective well-being. As families go nuclear and smaller, there is a growing need for closer social interaction among young people. The emergence of the internet provides convenience for modern people to develop and maintain social relations and meets the psychological needs of modern people for self-expression and social communication. Maintaining a rational and positive social attitude in the process of using the internet can regulate the use of the internet and establish a good interpersonal relationship, which can effectively reduce loneliness and mental illness. Sensitivity increased stress tolerance, met recognized psychological needs, and increased their overall life satisfaction. Therefore, young people will eventually trigger positive emotions in the social network of interpersonal relationships, improve social fit, increase interpersonal resources, and generate pleasant happiness through mutual support [36]. Therefore, social mentality helps young people gain social recognition, form the yearning and expectation for a better life, and finally further improve the subjective well-being of internet use. To sum up, we make the following assumptions:

**H10.** *Social mentality mediates between internet use and youth subjective well-being*.

**H11.** *Social mentality plays a moderating role between internet use and youth life satisfaction*.

The research model of this paper is shown in Figure 1.

## 3. Materials and Methods

### 3.1. Data Sources

This paper used the 2017 survey data of the China General Social Survey (CGSS) to examine the mechanism of physical exercise, internet use, life satisfaction, and social mentality on youth subjective well-being. CGSS started in 2003 and is the earliest national, comprehensive, and continuous academic survey project in China, carried out by the China Survey and Data Center of Renmin University of China. The latest round of surveys was implemented in 2017, covering 28 provincial-level administrative regions across the country. It has strong representation and reliability and is currently recognized as authoritative data. Due to the research theme, the samples were mainly selected from the cases that answered parts A and C, and the cases that answered “don’t know”, “unclear”, “refuse to answer”, “outliers”, and “missing” related to all variables in the paper were eliminated. Finally, a total of 619 valid samples were obtained. The sample characteristics are shown in Table 1.

### 3.2. Variable Measurement

#### 3.2.1. Explained Variable: Subjective Well-Being

Question D40 in the CGSS 2017 questionnaire was used to measure subjective well-being in this study. The questionnaire included 21 questions. This questionnaire is a six-point questionnaire, with responses including: 1 = strongly disagree; 2 = disagree; 3 = somewhat disagree; 4 = somewhat agree; 5 = agree; and 6 = strongly agree. The higher the total score, the greater the satisfaction with life.

#### 3.2.2. Explaining Variables: Physical Activity and Internet Use

Question A28 in the CGSS 2017 questionnaire was selected to measure internet use in this study. In the CGSS questionnaire, the main predictor is “How did you use the internet (including mobile) in the past year?” and the self-rating was recorded on a five-point scale as follows: 1 = never; 2 = rarely; 3 = sometimes; 4 = often; and 5 = very frequently. Higher scores indicate more physical activity.

Question A30 in the CGSS 2017 questionnaire was selected to measure physical exercise in this study. In the CGSS questionnaire, the main predictor is “In the past year, did you often participate in physical exercise?” and the self-rating was recorded on a five-point scale as follows: 1 = never; 2 = rarely; 3 = sometimes; 4 = often; and 5 = very frequently. Higher scores indicate more internet use.

#### 3.2.3. Mediating Variable: Life Satisfaction

Question D21 in the CGSS 2017 questionnaire was selected to measure life satisfaction in this study. In the CGSS questionnaire, the main predictor is “On the whole, are you satisfied with your living situation?” and the self-rating was recorded on a five-point scale as follows: 1 = very dissatisfied; 2 = dissatisfied; 3 = indifferent; 4 = satisfied; and 5 = very satisfied. Higher scores indicate higher levels of subjective well-being.

#### 3.2.4. Moderating Variables: Social Mindset

Based on previous research, this paper selected the three dimensions of social trust, social fairness, and happiness as dependent variables. Social trust corresponds to questions A33, A35, and A36 in the questionnaire. The higher the score, the better the social mentality.

#### 3.2.5. Control Variables

This paper used internal and external variables that may affect happiness, such as family, total income, age, gender, health status, and education, as control variables. Demographic variables such as education level, age, gender, and health status are essential variables affecting happiness [37].

### 3.3. Data Analysis

In this study, IBM SPSS Statistics version 24 was used to complete all data analysis and processing. We conducted a descriptive analysis to describe the basic sociodemographic characteristics of the study population, and a correlation analysis to verify the relationship between variables. In order to test whether there was mediating effect of life satisfaction and moderating effect of social mindedness, we used multivariate regression analyses and the SPSS macro PROCESS program (Model 4 and Model 8) [38]. A *p*-value of 0.05 was considered statistically significant. The study set bootstrap confidence interval (CI) at 95% based on 5000 bootstrapped samples. If zero was not included in the interval of 95% CI, it indicated that the mediation effect was significant.

## 4. Results

### 4.1. Descriptive Statistics and Correlation Analysis of Significant Variables

Before analyzing the relationship between physical exercise, internet use, life satisfaction, subjective well-being, and social mentality, the Pearson correlation coefficient between each variable was analyzed. Table 2 shows the correlation between the factors. Physical exercise was positively correlated with life satisfaction and subjective well-being, but not with social mentality. Internet use is positively correlated with life satisfaction and subjective well-being, but not with social mentality.

### 4.2. Effects of Physical Exercise and Internet Use on Youth Subjective Well-Being

Before analyzing the mediating effect of life satisfaction and the moderating effect of social mentality, this paper used the method of linear regression to analyze the simple effect of physical exercise and internet use on youth subjective well-being. The analysis results presented in Table 3 and Table 4 show that, under the premise of controlling for age, gender, education level, and individual health status, both physical exercise (β = 0.1145) and internet use (β = 0.1325) have an impact on youth subjective well-being. There is a positive effect (*p* < 0.05). Assuming that H1 and H2 hold, the more physical exercise and the more frequent internet use, the higher the subjective well-being of youth.

### 4.3. Effects of Physical Activity and Internet Use on Youth Subjective Well-Being: The Mediating Role of Life Satisfaction

The test results are shown in Table 3 and Table 4. Life satisfaction has a positive effect on youth subjective well-being. Physical exercise and internet use can significantly positively affect youth life satisfaction and subjective well-being. Hypothesis H3, Hypothesis H4, and Hypothesis H6 are verified. At the same time, it can be seen from Table 5 that the mediating effect of life satisfaction between physical exercise and youth subjective well-being is 0.0391, the 95% confidence interval is [0.0073, 0.0711], and the direct effect of physical exercise on youth subjective well-being is 0.0756, the confidence interval is [0.0406, 0.0032], and neither confidence interval contains 0. Therefore, it can be determined that there is a mediating effect between physical exercise and youth subjective well-being. Physical exercise improves subjective well-being by promoting life satisfaction, assuming that H5 is supported.

At the same time, it can be seen from Table 5 that the mediating effect of life satisfaction between internet use and youth subjective well-being is 0.0417, the 95% confidence interval is [0.0094, 0.0804], and the direct effect of internet use on youth subjective well-being is 0.0956, the confidence interval is [0.0105, 0.0225], and neither confidence interval contains 0. Therefore, it can be determined that there is a mediating effect between internet use and youth subjective well-being, and internet use can improve youth subjective well-being by promoting life satisfaction, assuming that H7 is supported.

### 4.4. Effects of Physical Exercise and Internet Use on Youth Subjective Well-Being: Moderating Effects of Social Mentality

First of all, the interaction between physical exercise and social mentality, as well as the interaction between internet use and social mentality, have significant positive effects on youth subjective well-being, indicating that social mentality plays a role in physical exercise and youth subjective well-being, internet use, and youth subjective well-being. The relationship between the two groups played a moderating role, verifying H8 and H9. It can be seen from Table 6 and Table 7 that with the improvement in social mentality, the positive impact of physical exercise and internet use on youth subjective well-being has gradually increased.

In conclusion, this paper verifies the positive effects of physical exercise and internet use on young people’s subjective well-being. Among them, life satisfaction played a mediating effect. Social mentality has a moderating effect between physical exercise and youth subjective well-being, internet use, and youth subjective well-being. The specific mechanism of action is shown in Figure 2.

## 5. Discussion

On the basis on sorting out related concepts and research results at home and abroad, this study used CGSS 2017 extensive sample data for analysis to explore the specific mechanism of physical exercise and internet use on youth subjective well-being. It verified the mediating role of life satisfaction and the moderating effect of social mentality.

### 5.1. The Direct Impact of Physical Exercise and Internet Use on Youth Subjective Well-Being

This study found that physical activity positively predicted subjective well-being, and the results obtained are consistent with previous research findings [39]. The positive effects of physical activity include better physical condition, positive psychological experiences, and harmonious interpersonal relationships. Youth can also experience related experiences after long-term participation in physical activity. Moreover, young people’s participation in physical exercise can divert the irritable and depressed emotional state caused by social stress, divert attention, and improve their living and working conditions, such as having a positive life attitude and maintaining a high-spirited and progressive mental state, and enhance subjective well-being. After participating in physical exercise, the individual’s subjective emotional feelings will inevitably affect the environment and ordinary life scene in terms of exercise through body movements and positions. Solomon’s “opposite-process effect” reveals a reasonable explanation for the effect of physical exercise on subjective well-being [40]. Brain exercises and other organ exercises related to emotional cognition generated by physical exercise change the inconsistency between physical and emotional feelings and promote the dynamic unity of physical and emotional feelings [41,42]. Therefore, during or after physical exercise, individuals will also have a particular emotional experience after experiencing the original emotion, such as changing the intensity of the negative emotional experience. Therefore, physical exercise can improve the subjective well-being of young people.

Moreover, the process theory states that individual happiness comes from life events or participation in life events. Especially when physical exercisers experience a state of well-being, their psychological experience is better. Research by Piqueras also found that individuals who regularly participated in physical exercise had a 1.3-fold increase in happiness compared to their non-participating peers [43]. When there are changes in physical, social, and psychological resources, individuals feel that their needs are met or expect to be met, and they will have a sense of satisfaction and happiness in life. Therefore, the promotion of subjective well-being by physical activity and exercise may be an experience of inner harmony through the internal and external perception and cognition of their state of existence during or after physical activity.

The rapid development of the internet is a significant test of young people’s ability development. This study found that young people’s internet use is positively correlated with subjective well-being [5]. In the research on the influence of media use behavior on subjective well-being, it is found that the connection path between media use and subjective well-being is reflected in three paths: use and satisfaction, social comparison and class imagination, social integration, and identity adjustment [44]. These three paths also apply to the internet as a new medium. In innovation and diffusion theory, members of society will adopt new media as a status symbol. The adoption and use of the internet and mobile phones affect the class status perception of members of society to varying degrees. The adoption of new media can reflect not only economic strength but also cultural capital. Therefore, internet use can reshape class or stratum, affecting people’s perception of happiness [45].

### 5.2. Mediating Effects of Life Satisfaction

Life satisfaction is an overall assessment of an individual’s life over a certain period on criteria set by the individual. Research shows that lifestyle behaviors, such as physical activity and internet use, can also be influenced.

The relationship between physical activity and life satisfaction among youth has received attention. The results of this study show that the more physical exercise individuals undertake, the higher their life satisfaction is, which validates previous studies [23]. Mills has theoretically expounded the physical and mental mechanism by which physical exercise affects mental health, manifested in that individuals who regularly participate in physical exercise have good physical and mental health [46]. Physical and mental health is the central premise of individual life satisfaction and has a predictive effect on life satisfaction. Therefore, physical exercise is an indispensable part of life. Individuals who exercise can enhance their physical fitness, improve their bad emotional state, and reduce their susceptibility to psychological symptoms, enhancing their positive cognition of life status and promoting life satisfaction. Likewise, physical exercise can help improve human cognitive function and quality of life. Regular participation in physical exercise is a necessary condition for young people to live a happy life.

At the same time, physical exercise is a physical activity aimed at developing the body, enhancing physical fitness, improving health, regulating the spirit, and enriching cultural life [24]. Studies have shown that participating in physical exercise can bring a higher quality of life and life satisfaction compared to those who do not participate in physical exercise. Participating in physical exercise can significantly enhance people’s physical fitness. Lawton has demonstrated that physical activity through physical activity can significantly improve people’s subjective well-being [47]. It can be seen that physical exercise is the main factor affecting the health and daily life activities of young people. Participating in physical exercise can promote physical and mental health, enhance self-confidence, help improve young people’s life satisfaction, and ultimately achieve a sense of happiness.

Internet use subtly impacts young people’s lifestyle, behavioral characteristics, and thinking habits. An important feature of young people’s internet use is the need for social interaction. Youth groups’ internet use has long gone beyond a single information search and transmission function. The widespread social network communication, photo and video sharing, and even shopping, work, and study software among youth groups all emphasize their network communication functions. Users can chat online, befriend each other, and have deeper interactions based on that. In terms of life satisfaction, people’s use of the internet may positively impact their life satisfaction; that is, the internet can improve people’s mental health, enrich their leisure life, and improve life through online chat interaction and entertainment and leisure. First, the internet provides more ways to communicate and interact with others. Online communication through the internet can improve people’s self-esteem, and reduce loneliness and depression, thereby effectively improving people’s mental health and improving people’s inner life satisfaction [48]. Secondly, with the popularization of the internet, people’s consumption patterns have undergone fundamental changes. They can conduct leisure activities through the internet without leaving home and increase leisure time and leisure consumption [49].

At the same time, social support mechanisms for the impact of internet use on well-being also received attention [50]. The social support mechanism believes that material support, psychological comfort, emotional belonging, and identity based on social interaction are indispensable factors in enhancing life satisfaction. Social networks play the role of social support through network resources. With these resources, the more problems an individual can solve in real life, the higher their subjective well-being.

Therefore, the impacts of the internet on individual well-being are numerous. As a modern way of life, the use of the internet expands the scope of people’s social interaction, enriches individual life experiences, and improves the efficiency of information dissemination. Compared with the traditional way of life, the internet can enhance the pleasant communication experience and satisfactory life experience among people through its advantages of high efficiency, convenience, and speed [51].

### 5.3. The Moderating Effect of Social Mentality

This study confirms that social mentality moderates the effects of physical exercise and internet use on life satisfaction and subjective well-being. The moderating effect of social mentality shows that with the improvement in the level of social mentality, the positive effects of physical exercise and internet use on life satisfaction and subjective well-being are strengthened. First, the social mentality is defined as a diffuse state of social mood, consisting of social–emotional tone, social consensus, and social values shared by social members, and in the background and basis for social members to compare, communicate and cooperate. The social mentality itself has the function of self-regulation, the function of social organization and condensation, and the function of social guidance. Social mindset has been recognized as a protective factor for adolescent development, including mental health. Studies have found that under the background of a positive social mentality, the growth mentality of individual thoughts, emotions, and behaviors has a potential role in buffering the externalized behaviors of young people [52]. The results of this study show that social mindsets among physical exercise, internet use, and SWB moderated the association between individual externalizing behaviors and psychological health. The magnitude of this association strengthened as the level of social mentality increased, supporting the cushioning effect of the regulator.

On the one hand, when social mentality is high, physical exercise has a higher impact on life satisfaction and subjective well-being. Among them, the social guiding function of high social mentality is to exert a positive group reference effect, and group polarization effect, which has a positive impact on the mentality of social individuals so that individual social groups can be satisfied with references, and life satisfaction can be improved. Moreover, individuals with an excellent social mentality can continuously adjust their state with social development and changes, enhancing the sense of gain and satisfaction. People’s subjective experience, emotional vitality, physical value, interpersonal perception, and predicament coping ability have been improved after physical exercise, and the subjective psychological experience is very close to the sense of life satisfaction. Therefore, under the blessing of a high social mentality, the effect of physical exercise on life satisfaction and subjective well-being is significantly strengthened.

On the other hand, when the social mentality is high, internet use also has a higher impact on life satisfaction and subjective well-being. The fragmentation, liberalization, interactivity, and other characteristics of internet communication not only publicize the personality of young people but also lure them into the plastic stage of thought and behavior. Positive changes in social mentality can help young people master and control the trend and development of the situation and master the discourse power of network ideology in crucial periods, sensitive topics, and significant events. At the same time, an excellent social mentality can channel young people’s emotions, respond to doubts, stabilize expectations, enable young people to replace emotional vent with rational thinking and introspective reflection instead of cold complaints, to prevent one-sided and extreme thinking, and habits. At the same time, the composition of social mentality also contains the component of social members compared with each other, referring to the group mechanism [53]. It is pointed out that individuals’ judgments about their living conditions are often formed by comparing with others.

Furthermore, this comparison does not arise out of thin air; it is only based on social interaction that we can understand each other and have a realistic basis for comparison. If the person they interact with is in a higher-class position and a better state of life, the individual is prone to frustration and low self-esteem, reducing subjective well-being. On the contrary, if the relationship partner is in a lower-class position and a poorer state of life, it is easy to generate self-satisfaction through comparison and then have a more positive perception of life and enhance subjective well-being [54].

Therefore, a good social mentality may guide young people to make reasonable comparisons, reasonably express social needs and emotional states in the virtual interpersonal network intertwined by the internet, strengthen the satisfaction of spiritual and material needs, and generate life satisfaction. It also promotes the continuous improvement in subjective well-being.

## 6. Research Shortcomings and Prospects

This study confirmed the relationship between physical exercise, internet use, life satisfaction, and subjective well-being, and explored the specific influencing mechanism. It is found that social mentality plays a moderating role in the influence of physical exercise and internet use on life satisfaction. However, this paper still has the following deficiencies, which can be supplemented and expanded in the subsequent research. First of all, as for the data characteristics, all the data selected in this paper are cross-sectional data, which cannot reveal the causal relationship. Meanwhile, all the variables are from the same database, which may cause homology bias. Therefore, the use of multi-time-series data from different sources can be considered in subsequent studies to enhance the credibility of the study.

## 7. Conclusions

Physical exercise and internet use can positively impact young people’s life satisfaction and subjective well-being, and the mediating role of life satisfaction is established. In addition, in terms of the moderating effect of social mentality, this study found that a positive social mentality can positively enhance the effect of physical exercise and internet use on life satisfaction and subjective well-being. It shows the importance of social mentality, the weathervane that affects social development.

## Figures and Tables

**Figure 1 ijerph-19-11201-f001:**
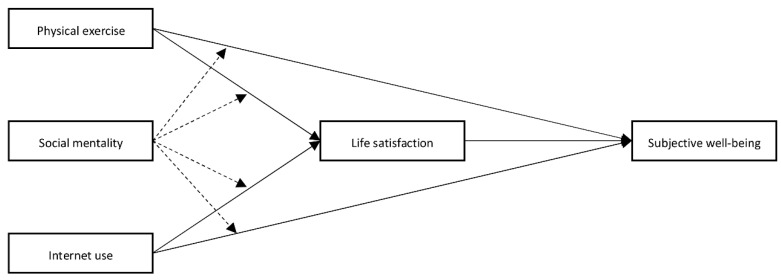
Hypothetical model diagram of physical activity and internet use on youth subjective well-being.

**Figure 2 ijerph-19-11201-f002:**
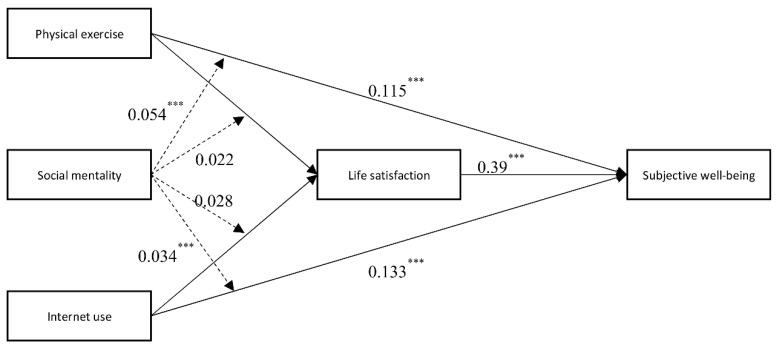
Mechanisms of physical exercise and internet use on youth subjective well-being. *** *p* < 0.001.

**Table 1 ijerph-19-11201-t001:** Sample characteristics.

Variable (619)	Percentage	Physical Exercise	Internet Use	Social Mentality	Subjective Well-Being
Educational level					
Junior high school and below	23.1	2.143	3.963	9.886	83.610
High school	23.7	2.913	4.585	9.535	86.680
College	17.8	2.935	4.705	10.325	88.660
Undergraduate	30.7	3.185	4.725	10.650	90.310
Graduate and above	4.7	3.340	4.520	10.620	93.410
Gender					
Male	43.9	3.000	4.600	9.850	87.368
Female	56.1	2.780	4.500	10.440	89.847

**Table 2 ijerph-19-11201-t002:** Descriptive statistics and correlation analysis of the main variables.

Variables	Mean	Standard Deviation	Sample Size	1	2	3	4
Physical exercise	2.881	1.352	619				
Internet use	4.544	0.756	619	0.126 **			
Social mentality	10.181	1.966	619	0.033	0.042		
Life satisfaction	2.253	0.826	619	0.093 *	0.110 **	0.324 **	
Subjective well-being	88.762	11.793	619	0.105 **	0.147 **	0.307 **	0.423 **

Note: * *p* < 0.05, ** *p* < 0.01.

**Table 3 ijerph-19-11201-t003:** Mediation model of life satisfaction on the effect of physical exercise on subjective well-being.

Result Variable	Predictor Variable	Fitting Index			Coefficient Significance	
		*R*	*R* ^2^	*F*	β	*t*
Subjective well-being		0.307	0.0943	12.5951 **		
	Age				0.0697	1.5367
	Gender				0.1034	2.6344 **
	Reading situation				0.0545	1.2155
	Health status				0.2543	6.5001 **
	Physical exercise				0.1145	2.8704 **
Life satisfaction		0.2557	0.0654	8.4681 **		
	Age				−0.0104	−0.2248
	Gender				0.2082	5.2200 **
	Reading situation				−0.0046	−0.1005
	Health status				0.1035	2.6047 **
	Physical exercise				0.0992	2.4476 *
Subjective well-being		0.4891	0.2392	31.6504 **		
	Age				0.0738	1.7732
	Gender				0.0214	0.5826
	Reading situation				0.0564	1.3689
	Health status				0.2136	5.9174 **
	Life satisfaction				0.3938	10.7264 **
	Physical exercise				0.0754	2.0517 *

Note: * *p* < 0.05, ** *p* < 0.01.

**Table 4 ijerph-19-11201-t004:** Mediation model of life satisfaction on the impact of internet use on subjective well-being.

Result Variable	Predictor Variable	Fitting Index			Coefficient Significance	
		*R*	*R* ^2^	*F*	β	*t*
Subjective well-being		0.315	0.0992	13.32521 **		
	Age				0.0515	1.1515
	Gender				0.1025	2.6225 **
	Reading situation				0.0441	0.9878
	Health status				0.2475	6.3179 **
	Internet use				0.1325	3.4045 **
Life satisfaction		0.2579	0.0665	8.6246 **		
	Age				−0.0262	−0.5755
	Gender				0.2067	5.1923 **
	Reading situation				−0.0133	−0.2931
	Health status				0.099	2.4826 *
	Internet use				0.1028	2.5941 **
Subjective well-being		0.4921	0.2422	32.1699 **		
	Age				0.0618	1.5037
	Gender				0.0217	0.5904
	Reading situation				0.0494	1.2032
	Health status				0.2087	5.7757 **
	Life satisfaction				0.3914	10.6749 **
	Internet use				0.0923	2.5686 *

Note: * *p* < 0.05, ** *p* < 0.01.

**Table 5 ijerph-19-11201-t005:** Main effect, direct effect, and mediating effect.

Relationship Path	Life Satisfaction	*Effect*	*SE*	*t*	95% CI		Effect Proportion
					*LL*	*UL*	
Physical exercise → Subjective well-being	Main effect	0.1148	0.04	2.87	0.0042	0.0362	100%
	Direct effect	0.0756	0.0369	2.0517	0.0406	0.0032	65.9%
	Mediation effect	0.0391	0.0165		0.0073	0.0711	34.1%
Internet use → Subjective well-being	Main effect	0.1373	0.0403	3.4045	0.0581	0.2164	100%
	Direct effect	0.0956	0.0372	2.5686	0.0105	0.0225	69.6%
	Mediation effect	0.0417	0.0181		0.0094	0.0804	30.4%

**Table 6 ijerph-19-11201-t006:** Test of the moderating effect of social mentality.

Variable Name	Subjective Well-Being				Life Satisfaction			
	*coeff*	*se*	*coeff*	*se*	*coeff*	*se*	*coeff*	*se*
Constant	−1.8466	0.3763	−1.6673	0.3736	−0.7526	0.427	−0.5698	0.3988
Independent variable								
Physical exercise	0.0753 *	0.036			0.0868	0.0385 *		
Internet use			0.0975 ***	0.0366			0.0952 *	0.0389
Mediation variable								
Life satisfaction	0.3401 ***	0.0379	0.3381 ***	0.0381				
Adjustment variable								
Social mentality	0.091 ***	0.0189	0.0896 ***	0.0189	0.1484	0.0194 ***		
Interaction item								
Physical exercise * social mentality	0.0538 ***	0.0179			0.0219	0.0191		
Internet use * social mentality			0.0337 *	0.0165			0.0284	0.0176
Control variable								
Age	0.0165	0.011	0.013	0.0109	−0.0081	0.0118	−0.0121	0.0116
Gender	0.0129	0.0728	0.014	0.0728	0.3319 ***	0.077	0.3278 ***	0.0767
Reading situation	0.0539	0.0336	0.0527	0.0337	0.0085	0.036	0.0058	0.036
Health status	0.2734 ***	0.0457	0.256 ***	0.046	0.1073 *	0.0489	0.0955 *	0.049
*R^2^*	0.276		0.2729		0.1491		0.1517	
*F*	28.6815		28.2423		15.1001		15.4053	

Note: * *p* < 0.05, *** *p* < 0.001.

**Table 7 ijerph-19-11201-t007:** Moderating effects of social mentality.

Variable	Social Mentality	Effect	BootSE	BootLLCI	BootULCI
Moderated mediating effect (physical exercise)	eff 1(M − 1SD)	0.0133	0.0231	−0.0319	0.059
	eff 2(M)	0.0282	0.0138	0.0008	0.0555
	eff 3(M + 1SD)	0.0431	0.0162	0.011	0.0753
Moderated mediating effect (internet use)	eff 1(M − 1SD)	0.0113	0.0215	−0.0214	0.0634
	eff 2(M)	0.0305	0.0157	0.0049	0.0659
	eff 3(M + 1SD)	0.0497	0.0226	0.0093	0.0991

## Data Availability

Publicly available datasets were analyzed in this study. These data can be found here: http://cnsda.ruc.edu.cn/index.php?r=projects/view&id=94525591 (accessed on 15 March 2022).

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
