# Peer review of "The Effect of Physical Exercise and Internet Use on Youth Subjective Well-Being—The Mediating Role of Life Satisfaction and the Moderating Effect of Social Mentality"

_ijerph, 2022, doi:10.3390/ijerph191811201_

Round 1
Reviewer 1 Report (New Reviewer)
This study is extremely interesting. On the one hand, it covers an existing gap in relation to the variables studied. On the other hand, it offers enlightening results for the proper use of the Internet and its possibilities in improving the well-being and health of young people, taking into account variables such as social mentality. It seems to me a very enriching and necessary work in the current scientific campus in relation to the health of our young people.The attached file indicates some suggestions or considerations to take into account to improve the clarity and rigor of your work.

Author Response
Please see the attachment.

Reviewer 2 Report (New Reviewer)
This manuscript could be an important reference for future studies. However, minor is still needed to improve the quality of this paper. Please revise the manuscript to address the expressed concerns. After thorough review, I am recommending some revisions. In this regard, kindly address the following comments and suggestions to further improve your manuscript
1. It was better if you wrote some of main finding as quantitative or mean ±SD within the abstract
2. Please write the type of study, sample size, sampling strategy and date and country of study in abstract
3. Introduction: The limitations of prior research might also be mentioned by the authors as further support for their present investigation.
4. write about all applied exclusion and inclusion criteria a bit more clearly by which you selected samples for this survey.
5. Mention the possible score (range) for each scales and meaning of it so easier to readers interpret the results
6. There are some spelling and grammatical errors in the text. Please correct them
7. You could increase the number of more recently studies in the reference section. You should have comprehensive and reliable comparisons between your findings with the other previous studies. Furthermore, write about the limitations of your survey. Are there any limitations for this study? If yes, please mention all limitations of current study within the discussion section, too.
8. Please mention the weak and strong points of your study
Author Response
Please see the attachment.

This manuscript is a resubmission of an earlier submission. The following is a list of the peer review reports and author responses from that submission.
Round 1
Reviewer 1 Report
It is not clear why the author decided to connect the two variables: Physical exercise and the use of Internet to discuss quality of life and well-being. How Physical exercise and the use of Internet are connected and affect quality of life? How their interconnections have a positive effect on youth subjective well-being ? The effect of physical exercise on well-being is a widely and recognized research trend, while the use of Internet can be a more original topic of interest if the author takes into account both risks and opportunities,the literature should be analysed in those terms.
the suggestion is to adjust the research trying to:
- find a valuable connection between the two variables
- explore risks and opportunity of the use of Internet.
Reviewer 2 Report
Thank you for allowing me to read a paper that has a significant impact on adolescents.
I was impressed with the effort put into the analysis with many analyses and many variables.
But, unfortunately, at the conceptual level, it is difficult to understand logically.
Why the Internet and the physical exercise?
These seem to be concrete elements compared to other concepts.
Regarding the structure of subjective well-being, it is necessary to consider other factors.
Subjective well-being and social spirit can be covariate.
The composition of each scale is unknown.
There are also various amounts and types of exercise. How do you organize it?
There is no indication of what to analyze.
Table 1 does not have the number of analytes.
For Internet usage, is the unit "time"?  Table 2
The results are mixed with analytical methods.
Is the version of SPSS used different each time?
I believe that the reason why it is difficult to understand the overall framework of the paper is the inadequacy of conceptual organization.
Opinions on the consideration of detailed results are difficult.
Be clear about the logical organization of concepts and why you have considered the relevance of exercise and the use of the Internet to the larger concepts of social spirit and subjective well-being.
I recommend that you focus on “4.3. The moderating effect of social mentality.”